# Use of a Novel Passive E-Nose to Monitor Fermentable Prebiotic Fiber Consumption

**DOI:** 10.3390/s25030797

**Published:** 2025-01-28

**Authors:** Lawrence Kosinski, Phillip A. Engen, Barbara Swanson, Michelle Villanueva, Maliha Shaikh, Stefan J. Green, Ankur Naqib, Bruce Hamaker, Thaisa M. Cantu-Jungles, Ali Keshavarzian

**Affiliations:** 1VOCnomics, LLC, Scottsdale, AZ 85255, USA; 2Rush Center for Integrated Microbiome and Chronobiology Research, Rush University Medical Center, Chicago, IL 60612, USA; phillip_engen@rush.edu (P.A.E.); barbara_a_swanson@rush.edu (B.S.); michelle_villanueva@rush.edu (M.V.); maliha_shaikh@rush.edu (M.S.); ankur_naqib@rush.edu (A.N.); ali_keshavarzian@rush.edu (A.K.); 3Rush University College of Nursing, Rush University Medical Center, Chicago, IL 60612, USA; 4Department of Internal Medicine, Rush University Medical Center, Chicago, IL 60612, USA; stefan_green@rush.edu; 5Genomics and Microbiome Core Facility, Rush University Medical Center, Chicago, IL 60612, USA; 6Whistler Center for Carbohydrate Research, Department of Food Science, Purdue University, West Lafayette, IN 47907, USA; hamakerb@purdue.edu (B.H.); tcantuju@purdue.edu (T.M.C.-J.); 7Department of Anatomy and Cell Biology, Rush University Medical Center, Chicago, IL 60612, USA; 8Department of Physiology, Rush University Medical Center, Chicago, IL 60612, USA

**Keywords:** electronic nose, microbiota modulation, prebiotics, volatile organic compounds

## Abstract

We developed a home-based electronic nose (E-Nose) to passively monitor volatile organic compounds (VOCs) emitted following bowel movements and assessed its validity by correlating the output with prebiotic fiber intake. Healthy, non-overweight participants followed a three-week protocol which included the following: (1) installing the E-Nose in their bathroom; (2) activating the device following each bowel movement; (3) recording their dietary intake; (4) consuming a fiber bar (RiteCarbs) containing a blend of 10 g of prebiotic fiber daily during weeks two and three; and (5) submit stool specimens at the beginning and end of the study for 16S rRNA gene sequencing and analysis. Participants’ fecal microbiome displayed significantly increased relative abundance of putative total SCFA-producing genera (*p* = 0.0323) [total acetate-producing genera (*p* = 0.0214), total butyrate-producing genera (*p* = 0.0131)] and decreased Gram-negative proinflammatory genera (*p* = 0.0468). Prebiotic intervention significantly increased the participants’ fiber intake (*p* = 0.0152), E-Nose Min/Max (*p* = 0.0339), and area over the curve in VOC–to–fiber output (*p* = 0.0044). Increased fiber intake was negatively associated (*R*^2^ = 0.53, *p* = 0.026) with decreased relative abundance of putative Gram-negative proinflammatory genera. This proof-of-concept study demonstrates that a prototype E-Nose can noninvasively detect a direct connection between fiber intake and VOC outputs in a home-based environment.

## 1. Introduction

The term “dietary fiber” encompasses plant carbohydrates that are resistant to human digestion. A higher intake of dietary fiber has been associated with reduced risk of multiple diseases including heart disease, colon cancer, diabetes, and Crohn’s disease [1,2]. Both soluble and insoluble fibers can be fermented by gut bacteria to different extents, producing short-chain fatty acids (SCFAs) with various health benefits [3]. While soluble fibers are generally more rapidly fermentable, insoluble fermentable fibers might have an important role in supporting important butyrogenic bacteria [4,5]. During fermentation, fiber is broken down by the gut microbiota, producing gases and volatile organic compounds (VOCs) like SCFAs [2]. These SCFAs (acetate, propionate, and butyrate) serve as nutrient sources and signaling molecules, helping to maintain barrier integrity and prevent systemic inflammation [6]. Furthermore, SCFAs can stimulate the release of glucagon-like Peptide-1 (GLP-1) from intestinal L-cells, a hormone associated with reduced appetite, increased insulin sensitivity, and improvements in parameters of metabolic syndrome [7].

The World Health Organization recommends an average daily intake of 25 g of fiber per adult [8]. This amount of fiber can be obtained by the daily consumption of vegetables, legumes, and whole grains. Assessment of dietary fiber intake typically relies on questionnaires like food frequency questionnaires and food records [9]. However, these methods do not assess fermentable fiber intake, which presents several challenges. First, there are no comprehensive databases that provide fermentability estimates for various food ingredients. Second, the fermentability of fiber can vary based on an individual’s unique gut microbiome. Acquisition of fermentation rate data from individuals relies on the direct measurement of VOCs, like SCFAs, that are sensitive to fermentable dietary fiber intake, but such measurements have historically been impractical to perform at home. While 90% of SCFAs are absorbed by the colonic mucosa [10], the remaining 10% are passed into the stool and can be measured in the headspace above a bowel movement [11]. Unfortunately, this assessment requires stool collection for gas chromatography analysis, limiting its widespread application. To overcome this limitation, we developed a modified electronic nose (E-Nose) capable of recording stool-derived VOCs emanating from a home toilet. To assess the performance of our modified E-Nose, we conducted an open-label pilot study to determine if our E-Nose could detect increased VOC production after 14 days of daily SCFA-promoting prebiotic bar consumption.

## 2. Materials and Methods

### 2.1. Study Design

A three-week-long prospective, open-label, proof-of-concept, observational study.

### 2.2. Participants

The study cohort consisted of 14 healthy individuals who were not morbidly overweight. After collecting the data and samples, 12 participants were evaluated for fiber intake and VOC output, while microbiota analysis was completed for 11 of them (Table 1). Participants were eligible if they (a) were between the age of 21 and 65, (b) had no chronic medical disease, with the exception of mild and controlled hypertension and hyperlipidemia, and (c) were willing to eat prebiotic bars, provide biological specimens, and answer questionnaires. Participants were excluded if they (a) had a body mass index (BMI) > 35 or <18, (b) regularly used nonsteroidal anti-inflammatory drugs (NSAIDs) for more than three days per week, (c) were diagnosed with a chronic gastrointestinal disorder (e.g., inflammatory bowel disease, irritable bowel syndrome requiring regular therapy, celiac disease, colon cancer, or a history of intestinal resection), (d) were allergic to almonds, flaxseed, or coconut, (e) followed a restricted diet (e.g., gluten-free, paleo, vegetarian, or vegan), (f) were unable to provide informed consent, or (g) lacked English-language proficiency. Individuals using fiber supplements or MiraLAX were allowed to participate in the study.

### 2.3. Prebiotic Supplement

Participants were given proprietary prebiotic fiber bars (RiteCarbs, West Lafayette, IN, USA) that we have previously shown to be highly fermentable and to promote SCFA production [12]. The bar contains the following fiber mixture: 30% resistant starch (raw potato starch), 30% resistant maltodextrin (Nutriose^TM^, Philadelphia, PA, USA), 30% stabilized rice bran, and 10% agave branched inulin.

#### Stool Collection

Stool samples were collected at two points in time: (1) baseline and (2) after 14 days of daily prebiotic use. Prior to their scheduled clinical visit, participants used an anaerobic home collection kit (BD GasPak, Becton Dickinson and Company, Sparks, MD, USA) to collect stool samples from the preceding 12–24 h. The samples were then transported to the laboratory and stored at −80 °C until analysis.

### 2.4. Electronic Nose

The E-Nose is a home-based device that contains a Bosch BME 680 VOC sensor, a Wi-Fi chip, internal fans, and buttons to identify specific users (Figure 1) [13]. The BME680 is the first gas sensor that integrates high-linearity and high-accuracy gas, pressure, humidity, and temperature sensors. It is especially developed for mobile applications and wearables where size and low power consumption are critical requirements. The BME680 guarantees—depending on the specific operating mode—optimized consumption, long-term stability, and high EMC robustness. In order to measure air quality for personal wellbeing, the gas sensor within the BME680 can detect a broad range of gases such as volatile organic compounds—see spec sheet (Table 2).

The device plugs into a bathroom AC receptacle within 8 feet of the commode and the internal fans draw ambient air over the sensor which contains a film of semiconducting metal oxides. The oxides react with gas molecules in the atmosphere which dissociate into charged ions or complexes that alter the resistance of the film. The change in resistance, measured in ohms, comprises the E-Nose output. The sensor output is then transmitted to a local Wi-Fi network and then to a cloud-based server for data storage and analysis.

### 2.5. Procedures and Questionnaires

Participants were instructed to: (a) plug the E-Nose into an AC receptacle in their bathroom within 8 feet of the commode and connect the device to their home Wi-Fi, (b) comply with user identification by pressing a button on the device after each bowel movement, (c) record their dietary food intake for the entire study period, and (d) consume their normal diet for three weeks. In weeks two and three (14 days), participants were instructed to (e) ingest one prebiotic fiber bar (RiteCarbs) containing 10 g of a highly fermentable fiber blend daily, and (f) submit two stool specimens for microbial community characterization using 16S rRNA gene amplicon sequencing, baseline analysis, and analysis following two weeks of prebiotic fiber bar supplementation.

At baseline, participants completed the Vioscreen questionnaire [14], a dietary assessment tool designed to evaluate their dietary habits over the past month, to assess inclusion criteria. Participants completed the NIH Patient-Reported Outcomes Measurement Information System (PROMIS GI) questionnaire to assess the impact of prebiotic intervention on gastrointestinal symptoms, including bowel movements, stool consistency, abdominal discomfort/pain, bloating, and flatulence (Table 1) [15].

### 2.6. Electronic Nose Volatile Organic Compound Output Data Interpretation

The E-Nose sends continuous output to the VOCnomics cloud-based server. The output reflects the resistance in ohms from the device, which is negatively affected by the presence of ions emitted into the air. VOCs emitted following a bowel movement are ionically charged and reduce resistance on the sensor. The resistance pattern begins with an abrupt drop, followed by a gradual return to baseline. We independently calculated the Min/Max fall, as well as the area over the curve, which was the product of Ohms times Time. A typical bowel movement pattern is shown in Figure 2.

### 2.7. DNA Extraction and Next-Generation Sequencing

Fecal samples underwent automated DNA extraction using a chemagic 360 instrument (Revvity, Shelton, CT, USA) and a chemagic DNA Stool 200 Kit H96, following the manufacturer’s instructions. Prior to purification on the chemagic instrument, fecal samples were bead-beaten using a TissueLyser II device (Qiagen, Hilden, Germany), as described previously [16]. The extracted genomic DNA was subsequently PCR-amplified using a two-stage protocol and employing primers targeting the V4 variable region of microbial 16S rRNA genes, as described in prior work [17]. The primers contained 5′ common sequence tags known as Fluidigm common sequences 1 and 2 (CS1 and CS2). For the first-stage amplifications, the study employed primers CS1_515F and CS2_806R. These primers were modified from the primer set used in the Earth Microbiome Project (EMP) and had the following sequences: CS1_515F—ACACTGACGACATGGTTCTACAGTGTGYCAGCMGCCGCGGTAA and CS2_806R—TACGGTAGCAGAGACTTGGTCTCCGGACTACNVGGGTWTCTAAT (with the underlined regions representing linker sequences). The PCR reactions were performed in 10 microliter volumes in 96-well plates, using repliQa HiFi ToughMix (Quantabio, Beverly, MA, USA). The PCR conditions were 98 °C for 2 min followed by 28 cycles of 98 °C for 10 s, 52 °C for 1 s, and 68 °C for 1 s.

For each sample, a second PCR amplification was performed in 10 μL reactions using repliQa HiFi ToughMix. Each well contained a unique 10-base barcode primer pair from the Access Array Barcode Library for Illumina. One μL of the first-stage PCR products was used as a template, without cleanup. The cycling conditions were 98 °C for 2 min followed by 8 cycles of 98 °C for 10 s, 60 °C for 1 s, and 68 °C for 1 s. The amplified libraries were pooled and sequenced with a 10% PhiX spike-in on an Illumina MiniSeq instrument using a mid-output flow cell (2 × 154 paired-end reads). All laboratory analyses were conducted by staff blinded to group assignment. DNA extraction, library preparation, and sequencing were conducted at the Genomics and Microbiome Core Facility (GMCF) at Rush University.

### 2.8. Bioinformatics and Statistical Analyses

Raw sequence data were processed using the QIIME2 software package (version 2023.5) [18]. Sequences were checked for quality with FastQC and merged using PEAR [19]. The merged sequences were quality-filtered using the q2-demux plugin, followed by denoising with DADA2 (via q2-dada2) [20]. Primer adapter sequences were then removed using the cutadapt algorithm [21]. Taxonomy was assigned using the q2-feature-classifier classify-sklearn naive Bayes taxonomy classifier against the SILVA 138 99% reference database [22,23]. The contaminant removal software, decontam [24], detected eight contaminants based on the prevalence of amplicon sequence variants (ASVs) in the reagent negative blank controls using default parameters (Appendix A). These ASVs were removed from the dataset prior to downstream analysis of microbial community structure. Datasets were rarefied to a depth of 16,500 sequences/sample for diversity analyses [25].

Microbial community structure was assessed through a series of standard analyses, including measures of within-sample diversity (alpha-diversity), between-sample diversity (beta-diversity), and identification of microbial features with altered relative abundance between groups. Specifically, alpha-diversity metrics (i.e., Shannon index, Simpson’s index, observed features (richness), and Pielou’s evenness) and beta-diversity metrics were calculated using q2-diversity within the QIIME environment. To compare microbial community structures between groups, permutational multivariate analysis of variance (PERMANOVA) and permutational analysis of multivariate dispersions (PERMDISP; 9999 permutations), both based on Aitchinson distance, were performed [26,27]. Visualization of distance data was performed using centroid-based non-metric multi-dimensional scaling (NMDS) plots for all metadata groups using the vegan package in R. The Wilcoxon signed-rank test for paired comparisons was used to identify significant differentially abundant features (individual taxa, inferred functional pathways) between participants’ baseline and prebiotic intervention. Inferred metagenomic functional pathways were assessed using the phylogenetic investigation of communities by reconstruction of unobserved states (PICRUSt2) plugin within the QIIME2 environment [28]. Pathways were annotated against the MetaCyc Metabolic Pathway Database [29]. Microbial features with relative abundances below 0.1% were ignored for differential abundance calculations. The Benjamini–Hochberg (BH) method was used to correct *p*-values for multiple testing. A random forest algorithm (Boruta) was employed to detect microbial genera features of importance between participants’ baseline and prebiotic intervention [30]. We did not have the necessary data to apply machine learning to the VOC output data. Based on surveys of relevant literature [31,32,33], microbial genera were categorized as putative SCFA-producing taxa or proinflammatory taxa. The relative abundances of these taxa were combined within each sample to create SCFA or proinflammatory aggregate values.

Descriptive statistics, clinical paired t-tests, and linear regression were performed as appropriate using GraphPad Prism 10.0 (GraphPad Software, San Diego, CA, USA). We conducted a linear regression analysis on each VOC output and the relative abundance outcomes of curated genera to examine the relationship between these two distinct out-comes. Due to the small sample size, we did not adjust the linear regression analysis for potential confounders, such as age, diet, or baseline microbiota.

## 3. Results

### 3.1. Prebiotic Fiber Intake and Electronic Nose Performance

Prebiotic bar consumption significantly increased daily fiber intake during the two-week supplementation period from baseline levels (*p* = 0.015) (Figure 3a). The E-Nose VOC data also indicated a response to the increased fiber intake. Both the total VOC output (*p* = 0.033) and the VOC–to–fiber ratio (*p* = 0.044) were significantly elevated in participants following prebiotic intervention. (Figure 3b,c). The VOC data are provided in Appendix A.

### 3.2. Prebiotic Intervention Effects on Gut Microbiota

A machine-learning approach (random forest) identified significant genus-level microbial changes between baseline and post-treatment fecal samples (Appendix A). The prebiotic intervention was associated with changes in the fecal microbiome, including a significant increase in the relative abundance of putative total SCFA-producing genera (*p* = 0.032), and putative acetate (*p* = 0.021) and butyrate (*p* = 0.013) producers (Figure 4a–c). The relative abundance of putative propionate producers was not significantly different between baseline and post-treatment samples (*p* = 0.174) (Figure 4d). Additionally, the intervention was associated with a decrease in the relative abundance of Gram-negative bacteria (*p* = 0.046) and the ratio of Gram-negative–to–total SCFA-producing genera (*p* = 0.032) (Figure 4e,f). The Firmicutes–to–Bacteroides ratio trended higher post-intervention relative to the baseline (*p* = 0.053) (Figure 4g). Linear regression analysis revealed a significant association (*R*^2^ = 0.53, *p* = 0.026) between increased fiber intake and a reduction in the relative abundance of putative Gram-negative proinflammatory bacterial genera (Figure 4h).

Using non-targeted differential abundance analyses, intervention was associated with changes in the relative abundances of several microbial phyla and genera (Appendix A). Notably, the relative abundance of bacteria from the phylum Desulfobacterota (sulfate-reducing microorganisms) decreased following prebiotic intervention (*p* = 0.042; *q* = 0.296). Similarly, the relative abundance of the putatively proinflammatory genus *Bilophila* (Proteobacteria) was also reduced after prebiotic intervention (*p* = 0.034; *q* = 1.00) (Appendix A). Conversely, the intervention led to a trend of increased relative abundance of putative beneficial bacteria from the genus *Fusicatenibacter* (*p* = 0.075; *q* = 1.00) and the genus-level taxon *Lachnospiraceae UCG-004* (*p* = 0.075; *q* = 1.00) (Appendix A). Inferred metagenomics functional pathway analysis suggested trending differences from prebiotics. This trend led to a decrease in the relative abundance of genes in the superpathway of L-methionine biosynthesis via sulfhydrylation, which may promote or alleviate gut inflammation (*p* = 0.147; *q* = 0.806) (27). Additionally, the trend caused an increase in the relative abundance of genes involved in sucrose degradation III (invertase) (*p* = 0.174; *q* = 0.806) and IV (sucrose phosphorylase) (*p* = 0.101; *q* = 0.806), associated with microbial fermentation and SCFA production in the gut (27). Although specific changes in microbial taxa were observed after intervention, the prebiotic intervention did not significantly impact the overall microbial community structure. This was evidenced by no significant change in alpha-diversity indices (Appendix A) or beta-diversity, as assessed through PERMANOVA (*p* = 0.999) and PERMDISP (*p* = 0.995) (Appendix A). The microbiota data are provided in Appendix A.

## 4. Discussion

In this pilot proof-of-concept study, we showed that our novel, home-based E-Nose detected favorable VOC changes following two weeks of SCFA-promoting prebiotic fiber consumption concordant with microbiota shifts, as measured by 16S rRNA gene amplicon sequencing and analysis. Our findings suggest the E-Nose holds potential for the in-home monitoring of adherence and response to microbiota-directed therapies. Additionally, the findings support follow-up investigations of the E-Nose for use in in-home gut-dysbiosis testing for at risk-populations, which could expand access to precision therapies.

The health benefits of dietary fiber have been well-described [34] and include, but are not limited to, the promotion of normal gut motility [35], control of body weight and visceral adiposity [36], improvement in insulin sensitivity [37], idealization of gut microbial flora (microbiota community) [2], control of chronic inflammation [38], prevention of cardiovascular disease [39], and colon cancer [40].

Despite an adequate understanding of the benefits of fiber intake and the belief Americans consume an adequate amount [41], national consumption surveys indicate that only about 5% of Americans meet fiber intake recommendations; and most would need to increase their fiber intake by 50% to reach compliance [42]. The reasons for this noncompliance are multiple, including taste and cultural tendencies, along with the transformation of the Western diet food environment to one inundated with highly processed food engineered to contain artificially high levels of sugar and fat [43]. The fiber content in these highly processed foods tends to be lower than in meals prepared from raw ingredients [44]. Thus, well-balanced fiber supplements that are culturally acceptable and tolerable with minimal to no side effects should be considered to overcome this unmet dietary need in most of the United States population.

In addition to the above issues, product labeling only includes an assessment of total fiber content [45], which fails to capture the balance of fermentable and non-fermentable fiber. The fermentability of fiber is a key factor in determining the scope of its health benefits, influencing microbial diversity and function in the gastrointestinal tract [46,47]. The disease-ameliorating ability of dietary fiber is often attributed to SCFAs produced from fermentable fiber breakdown, which serves as a nutrient source and in signaling molecules both within the intestinal tract and throughout the body.

Monitoring of daily fermentable fiber intake is yet another challenge to ensure individuals have achieved the recommended, health-promoting intake of daily fiber. Our proof-of-concept pilot study suggests that passive E-Nose monitoring and intake of a tolerable, balanced prebiotic mixture can help overcome these obstacles to achieve the stated goal. Noteworthy in the field of obesity management are the relationship between SCFAs and the endogenous production of GLP-1 [7] and the ability for our prebiotic fiber to significantly increase SCFA-producing bacteria. Although GLP-1 is mainly synthesized and secreted by enteroendocrine L-cells of the gastrointestinal tract [7], this secretion is partly mediated by the direct nutrient sensing of G-protein-coupled receptors, which specifically bind to monosaccharides, peptides and amino-acids, monounsaturated and polyunsaturated fatty acids, as well as to short-chain fatty acids. Foods rich in these nutrients, such as high-fiber grain products, nuts, avocados, and eggs, also seem to influence GLP-1 secretion and could promote associated beneficial outcomes in healthy individuals as well as individuals with type 2 diabetes or with other metabolic disturbances [7].

We acknowledge several limitations in this pilot, proof-of-concept, study. The most significant is the reliance on participant self-reporting to measure dietary intake. Our coordinators worked closely with the participants to obtain accurate reporting, but we cannot rule out recall/reporting errors. A second limitation is related to the non-standardized environment of each bathroom. Some bathrooms contained intermittently activated ceiling exhaust fans that could have affected VOC output surrounding bowel movements. A third limitation was the small sample size that limited statistical power and precluded adjusting our analyses for potential covariates of microbiota response and VOC production, including age, BMI, and baseline diet. Other limitations were the relatively small sample size being predominately female, the short duration of the study, the open-label study design, and absence of a placebo control condition.

Our findings can inform the design of future hypothesis-testing studies. These studies could include the use of a second-generation multi-channel E-Nose designed to enhance sensor precision and multi-VOC profiling. An upgraded device holds the potential to distinguish specific VOCs associated with microbial taxa and their metabolic pathways, thus increasing microbiota resolution. We also recommend future investigations that concurrently measure VOCs and parameters of metagenomics, metabolomics, transcriptomics, and machine learning to explicate a multidimensional understanding of interactions between the gut microbiome and dietary fiber intake, and to facilitate deeper mechanistic insight [48].

Another, and underexplored, approach, involves photonics-based colorimetric chemosensors to measure VOCs [49,50]. One device that incorporates this approach is the optoelectronic nose, which uses chemically responsive pigments printed on disposable porous polymer membranes or paper. The sensor array responds to a wide range of chemical reactivity parameters, as opposed to the more limited physical reactions, such as adsorption, detected by metal oxide sensors [51]. This approach also holds the potential to address the “sensor drift” associated with metal oxide-based sensors that can affect reproducibility and accuracy [52]. To date, most optoelectronic nose studies have investigated environmental volatiles (e.g., explosive detection), with a growing number investigating signatures associated with pathogenic microbes and diseases [53,54]. Future studies comparing the home-based E-Nose device with photonics-based colorimetric chemosensors to measure VOCs are required.

## 5. Conclusions

In summary, our findings support proof-of-concept that the E-Nose can detect changes in VOCs following a two-week trial of an SCFA-promoting prebiotic supplement. These findings suggest the device holds potential as the first technology to passively monitor fermentable fiber intake at home, and thus, objectively monitor and document adherence to fermentable fiber supplements. This technology could directly address methodological and logistical issues that have limited fiber-adherence monitoring, especially in clinical trials. Additional studies, adequately powered for hypothesis testing and including placebo controls, are needed to explicate the validity of this device for identifying individuals with obesity-related intestinal dysbiosis, measuring adherence to microbiota-directed therapies, and distinguishing responders from non-responders.

## 6. Patents

The electronic nose is owned by VOCnomics, LLC. A nonprovisional patent is pending (patent applicant: VOCnomics, LLC; inventor: Lawrence R Kosinski, MD; application number: PCT/US23/70684; status: pending; title: METHOD AND APPARATUS FOR REMOTE MONITORING OF VARIOUS ORGANIC COMPOUNDS). The prebiotic bar was developed by RiteCarbs, LLC, co-owned by Drs. Keshavarzian, Hamaker, and Cantu-Jungles. A provisional patent is pending for the prebiotic mixture used in the bar (patent applicant: Purdue Research Foundation; inventors: Drs. Keshavarzian, Hamaker, and Cantu-Jungles; application number: 18579679; status: pending). The patent covers the use of prebiotics to improve health.

## Figures and Tables

**Figure 1 sensors-25-00797-f001:**
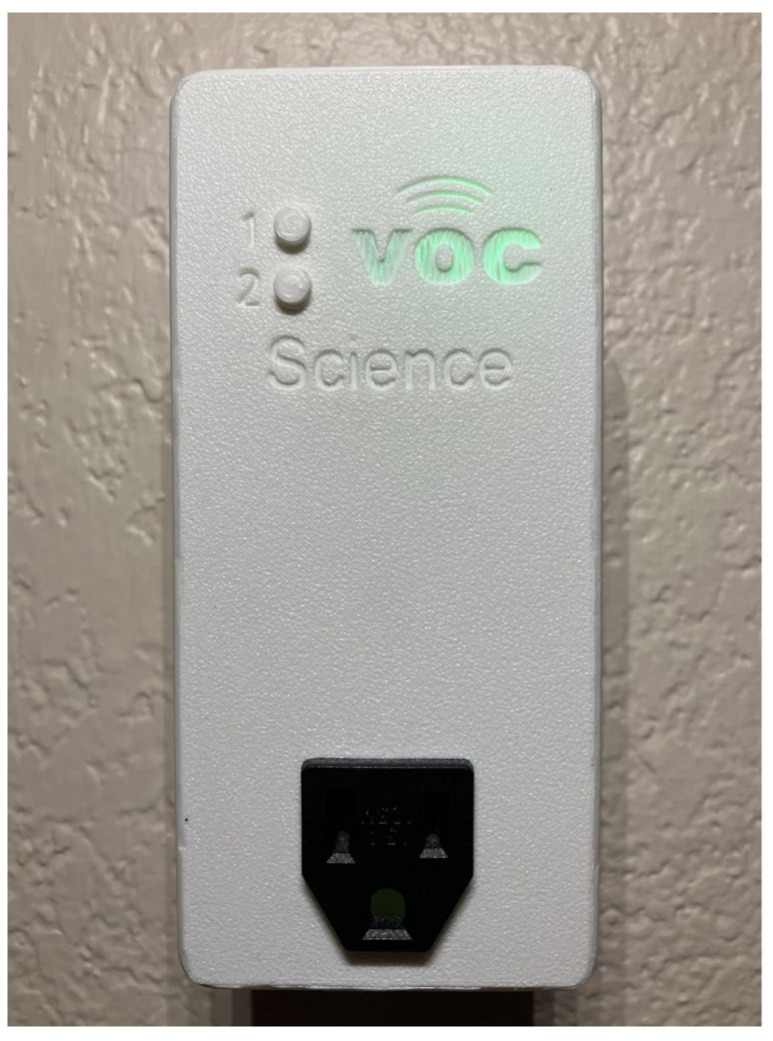
Electronic nose device. The E-Nose device features a Bosch BME 680 VOC sensor and includes an internal fan to draw air across the sensor. It is equipped with buttons to identify different users and has a built-in Wi-Fi transmitter for communication. The device’s output is transmitted and stored on a cloud-based server.

**Figure 2 sensors-25-00797-f002:**
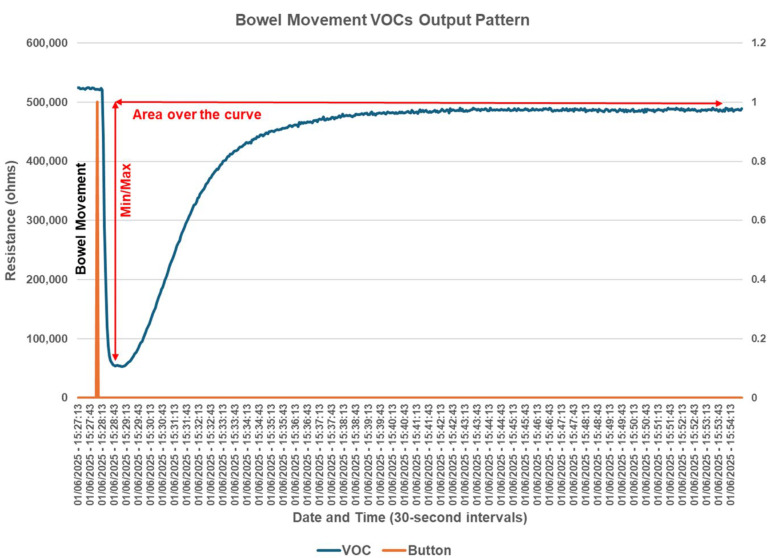
Volatile organic compound output following a bowel movement. This is an example of a typical bowel movement pattern depicting how both the Min/Max fall values and area over the curve data were generated for each individual research participant.

**Figure 3 sensors-25-00797-f003:**
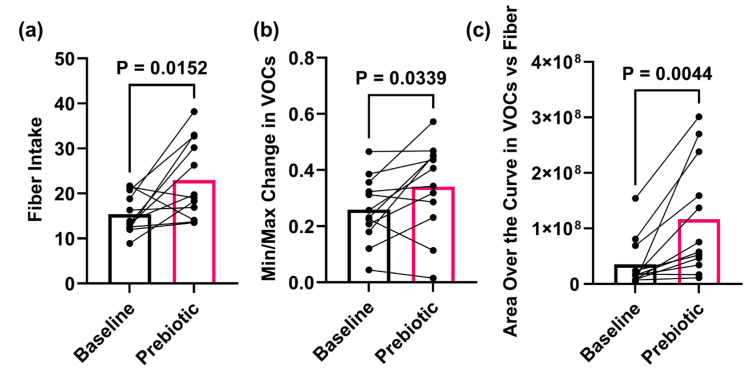
Fiber intake and electronic nose performance. (**a**) The participants’ fiber intake notably increased after two weeks of prebiotic intervention (baseline: 15.42 ± 4.27; prebiotic: 22.96 ± 8.68). As a result of this increased fiber intake, both the E-Nose (**b**) Min/Max change in VOCs output (baseline: 0.25 ± 0.11; prebiotic: 0.34 ± 0.15) and (**c**) the area over the curve VOCs–to–fiber ratio (baseline: 35,066,405 ± 44,835,904; prebiotic: 116,642,403 ± 102,906,142) were higher in the participants. Statistical analyses: A line connects each participant at baseline and after the prebiotic intervention. Bar height represents the group’s mean value and individual samples are indicated. *n* = 12. Wilcoxon signed-rank paired test. These data are provided in Appendix A.

**Figure 4 sensors-25-00797-f004:**
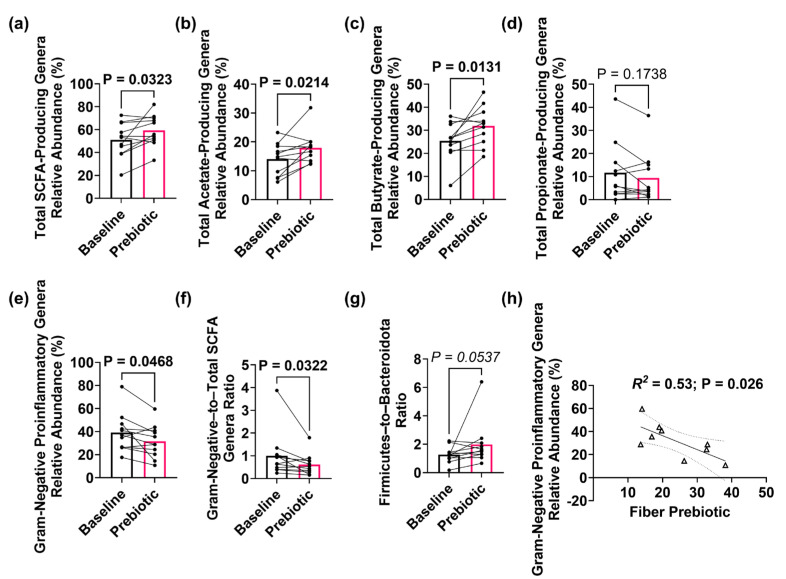
Prebiotic consumption beneficially impacted the microbiota. Analysis of curated genera lists revealed that the relative abundances of (**a**) total SCFA-producing taxa, (**b**) total acetate-producing taxa, and (**c**) total butyrate-producing taxa were all significantly higher following prebiotic intervention. (**d**) Total propionate-producing taxa relative abundance remained unchanged with prebiotics. In contrast, (**e**) the relative abundance of Gram-negative proinflammatory taxa and (**f**) Gram-negative–to–total SCFA ratio were both significantly lower post-intervention. (**g**) The Firmicutes–to–Bacteroidota ratio trended higher. (**h**) A significant relationship was observed between the reduction in Gram-negative proinflammatory genera and increased fiber intake following the two-week prebiotic intervention. Statistical analyses: A line connects each participant at baseline and after prebiotic intervention. Bar height represents the group’s mean value and individual samples are indicated. *n* = 11. (**a**–**g**) Wilcoxon signed-rank paired test and (**h**) linear regression. The microbiota data are provided in Appendix A.

**Table 1 sensors-25-00797-t001:** Participants’ demographics.

Characteristics	*n* = 14	*n* = 12 ^+^	*n* = 11 *
Age (mean ± SEM, Min/Max range)	39.29 ± 3.79 (22–58)	38.42 ± 4.12 (22–58)	37.09 ± 3.92 (22–57)
BMI (mean ± SEM)	26.73 ± 0.87	26.36 ± 0.92	25.98 ± 0.96
Sex			
Female (*n*,%)	12 (85.7%)	11 (91.7%)	10 (90.9%)
Male (*n*,%)	2 (14.3%)	1 (8.3%)	1 (9.1%)
Race			
Asian (*n*,%)	2 (14.3%)	2 (16.7%)	2 (18.2%)
Caucasian (*n*,%)	12 (85.7%)	10 (83.3%)	9 (81.8%)
Ethnicity			
Not Hispanic or Latino (*n*,%)	6 (42.9%)	5 (41.7%)	5 (45.5%)
Hispanic or Latino (*n*,%)	8 (57.1%)	7 (58.3%)	6 (54.5%)

Gastrointestinal symptoms were assessed via the PROMIS gastrointestinal symptom scale. None of the 33 PROMIS gastrointestinal outcome measures showed a significant difference across all participants, indicating that none of the participants experienced notable gastrointestinal symptoms while consuming the prebiotic bar for 14 days. ^+^ Demographic information of the participants analyzed for fiber intake VOC outputs. * Demographic information of the participants analyzed for fecal microbiota. Medications: one participant was on medication for high blood pressure and a statin, one participant took allergy medications (Singular and Claritin), one participant was on high blood pressure medication, and one participant was taking a statin. *n* = total number of participants examined; Age = Average age of the group, along with the standard error of mean (SEM), and the minimum and maximum ages recorded in the dataset; Body Mass Index (BMI) = Average BMI of the group, along with the standard error mean (SEM) of the BMIs recorded in the dataset; Sex = total number (*n*) and percentage (%) of female or male participants examined; Race = total number (*n*) and percentage (%) of Asian or Caucasian participants examined; Ethnicity = total number (*n*) and percentage (%) of Not Hispanic or Latino or Hispanic or Latino participants examined.

**Table 2 sensors-25-00797-t002:** Bosch BME 680 VOC Sensor Technical Specification Sheet.

Parameter	Technical Data
Package Dimensions	8-Pin LGA with metal 3.0 × 3.0 × 0.93 mm^3^
Operation Range (Full Accuracy)	Pressure: 300–1100 hPa Humidity: 0–100% Temperature: −40–85 °C
Supply Voltage VDDIO	1.2–3.6 V
Supply Voltage VDD	1.71–3.6 V
Interface	I^2^C and SPI
Average Current Consumption (1 Hz data refresh rate) Average Current Consumption in Sleep Mode	2.1 µA at 1 Hz Humidity and Temperature 3.1 µA at 1 Hz Pressure and Temperature 3.7 µA at 1 Hz Humidity, Pressure and Temperature 0.09–12 mA for p/h/t/gas, depending on operation mode
Gas Sensor	
Response Time (_T_ 33–63%)	<1 s (for new sensors)
Sensor–to–Sensor Deviation	±15% ± 15
Power Consumption	<0.1 mA in Ultra-Low Power Mode
Output Data Processing	Direct Output of Index of Air Quality (IAQ)
Humidity Sensor	
Response Time (_T_ 0–63%)	8 s
Accuracy Tolerance	±3% Relative Humidity
Hysteresis	≤1.5% Relative Humidity
Pressure Sensor	
RMS Noise	0.12 Pa (equiv. to 1.7 cm)
Sensitivity Error	±0.25% (equiv. to 1 m at 400 m height change)
Temperature Coefficient Offset	±1.3 Pa/K (equiv. to ±10.9 cm at 1 °C temperature change)

LGA = Land Grid Array; hPA = hectopascal; % = percent; V = volts; I^2^C = two-wire interface; SPI = Serial Peripheral Interface; Hz = hertz; p/h/t = pressure/humidity/temperature; _T_ = time constant; and s = seconds.

## Data Availability

The original contributions presented in this study are included in the Appendix A. Further inquiries can be directed to the corresponding author. The original data presented in the study are openly available in the National Center for Biotechnology Information (NCBI) BioProject database under accession number PRJNA1179953 (16S rRNA). The SILVA 16S rRNA database used for alignment is available at https://data.qiime2.org/2023.5/common/silva-138-99-nb-classifier.qza (accessed on 10 February 2024).

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
