# Peer review of "Use of a Novel Passive E-Nose to Monitor Fermentable Prebiotic Fiber Consumption"

_sensors, 2025, doi:10.3390/s25030797_

Round 1

Reviewer 1 Report

Comments and Suggestions for Authors

This paper aimed to apply a prototype Electronic-Nose (E-nose) to noninvasively detect a direct connection between fiber intake and VOC outputs in a home-based environment. The topic is interesting. The experiments are well carried out, and the presentation and structure are acceptable. Taken together, the quality of this paper justifies its publication in Sensors.

Author Response

We are incredibly grateful to the reviewers and the editor for providing us with constructive and consistent feedback and for giving us the opportunity to revise and improve our manuscript entitled “Use of a Novel Passive E-Nose to Monitor Fermentable Prebiotic Fiber Consumption.

We have made modifications throughout the manuscript, which we believe addresses all the reviewers’ concerns. We have included detailed point-by-point responses to each of the reviewers' and editor’s concerns below. Our responses have been highlighted in blue to facilitate the review process. We have included a revised version of the manuscript with tracked changes (to ease the review process by the reviewers) and one without tracked changes. We believe that our manuscript has been significantly improved by the reviewers’ constructive comments.

Reviewer #1

Reviewer General Comment: “This paper aimed to apply a prototype Electronic-Nose (E-nose) to noninvasively detect a direct connection between fiber intake and VOC outputs in a home-based environment. The topic is interesting. The experiments are well carried out, and the presentation and structure are acceptable. Taken together, the quality of this paper justifies its publication in Sensors.”

Response: We sincerely thank Reviewer 1 for their positive comments and support for the publication.

Reviewer 2 Report

Comments and Suggestions for Authors

The manuscript describes a study evaluating a novel Electronic-Nose (E-Nose) device designed for home use to monitor volatile organic compounds (VOCs) emitted during bowel movements. The study's main story revolves around assessing the device's capability to correlate changes in VOC outputs with dietary fiber intake, specifically focusing on short-chain fatty acid (SCFA)-producing microbiota changes due to prebiotic fiber supplementation. However, while the study offers great potential, several key issues need to be addressed to strengthen the study. Therefore, I recommend a major revision of this manuscript before it is considered for publication.

The detailed comments are enumerated as follows:

1.       The format of the Abstract is very strange. Does it meet the journal's standards?

2.       The authors mention the development of a home-based Electronic-Nose (E-Nose); however, the mechanism of action for this E-Nose is not described in sufficient detail. It appears that the system might rely on a commercial sensor. If the E-Nose system is self-developed by the authors' team, the manuscript should include a detailed explanation of its design, sensor configuration, data acquisition process, and operational principles to ensure clarity and reproducibility for readers.

3.       The study lacks a control group (e.g., participants who did not consume the fiber supplement). Including a control group would strengthen the conclusions by isolating the effects of fiber supplementation from other confounding factors.

4.       The authors report correlations between E-Nose VOC outputs and microbiota changes but do not provide sufficient statistical evidence or detail on the correlation methods used. Were these correlations adjusted for potential confounders such as age, diet, or baseline microbiota composition?

5.       The manuscript mentions the use of machine learning for VOC pattern recognition, but the details of model validation (e.g., cross-validation, and accuracy metrics) are missing. This information is necessary to evaluate the reliability of the results. I suggest the authors have some literature review (e.g., Sens. Actuators B: Chem. 2022, 351, 130915) or comparison regarding the pattern recognition part.

Author Response

We are incredibly grateful to the reviewers and the editor for providing us with constructive and consistent feedback and for giving us the opportunity to revise and improve our manuscript entitled “Use of a Novel Passive E-Nose to Monitor Fermentable Prebiotic Fiber Consumption.

We have made modifications throughout the manuscript, which we believe addresses all the reviewers’ concerns. We have included detailed point-by-point responses to each of the reviewers' and editor’s concerns below. Our responses have been highlighted in blue to facilitate the review process. We have included a revised version of the manuscript with tracked changes (to ease the review process by the reviewers) and one without tracked changes. We believe that our manuscript has been significantly improved by the reviewers’ constructive comments.

Reviewer #2

Reviewer General Comment: “The manuscript describes a study evaluating a novel Electronic-Nose (E-Nose) device designed for home use to monitor volatile organic compounds (VOCs) emitted during bowel movements. The study's main story revolves around assessing the device's capability to correlate changes in VOC outputs with dietary fiber intake, specifically focusing on short-chain fatty acid (SCFA)-producing microbiota changes due to prebiotic fiber supplementation. However, while the study offers great potential, several key issues need to be addressed to strengthen the study. Therefore, I recommend a major revision of this manuscript before it is considered for publication.”

R.2.1 “The format of the Abstract is very strange. Does it meet the journal's standards?”

Response: The abstract has been reformatted to align with the journal’s standards. Thank you for bringing this to our attention.

R.2.2.  “The authors mention the development of a home-based Electronic-Nose (E-Nose); however, the mechanism of action for this E-Nose is not described in sufficient detail. It appears that the system might rely on a commercial sensor. If the E-Nose system is self-developed by the authors' team, the manuscript should include a detailed explanation of its design, sensor configuration, data acquisition process, and operational principles to ensure clarity and reproducibility for readers.

Response: Thank you for your suggestion. We have added the requested information on lines 118-141, along with a new Table 2 (Technical Specification Sheet), to provide a clearer description of the home-based Electronic-Nose.

R2.3. “The study lacks a control group (e.g., participants who did not consume the fiber supplement). Including a control group would strengthen the conclusions by isolating the effects of fiber supplementation from other confounding factors.

Response: We have acknowledged this in the limitations section of the Discussion (lines 359-361), stating that this pilot proof-of-concept provides a foundation for future studies that include controls and a larger sample size.

R2.4. “The authors report correlations between E-Nose VOC outputs and microbiota changes but do not provide sufficient statistical evidence or detail on the correlation methods used. Were these correlations adjusted for potential confounders such as age, diet, or baseline microbiota composition?”

Response: We conducted linear regression analysis on each VOC output and the relative abundance outcomes of curated genera to examine the relationships between these distinct outcomes. Due to the small sample size, we did not adjust the linear regression analysis for potential confounders such as age, diet, or baseline microbiota. This information has been added to the Methods section (lines 237-240). Additionally, in the Discussion section, we included a statement in the limitations paragraph (lines 357-359): "A third limitation was the small sample size, which limited statistical power and prevented us from adjusting our analyses for potential covariates of microbiota response and VOC production, such as age, BMI, and baseline diet."

Please note that the participants in this pilot study had similar baseline diet, gastrointestinal outcomes, age, and BMI—factors that could influence VOC production and response to fiber supplementation—which likely had no significant impact on our results.

R.2.5. “The manuscript mentions the use of machine learning for VOC pattern recognition, but the details of model validation (e.g., cross-validation, and accuracy metrics) are missing. This information is necessary to evaluate the reliability of the results. I suggest the authors have some literature review (e.g., Sens. Actuators B: Chem. 2022, 351, 130915) or comparison regarding the pattern recognition part.”

Response: We apologize for not providing a clearer definition of the machine learning methods. While we did not apply machine learning to VOCs, we used it on microbial genera that exhibited compositional changes before and after prebiotic intervention (lines 229-231). We do not have data to perform machine learning on VOCs, and we will acknowledge this for future investigations to measure (lines 366-370).

Reviewer 3 Report

Comments and Suggestions for Authors

This paper investigates a home-based Electronic-Nose (E-Nose) to passively monitor volatile organic compounds (VOCs) emi>ed with a bowel movement and assessed its validity by correlating the output with prebiotic fiber intake. Here are some suggestions.

1.     Enhance Sensor Precision and Multi-VOC Profiling.Upgrade the E-Nose with multi-channel sensing technology to distinguish between specific volatile organic compounds (VOCs) associated with diverse microbial metabolic pathways. This would allow precise correlation of VOC profiles with microbial taxa and functional pathways, thus increasing the resolution of microbiota monitoring.

2.     Incorporate Omics Data for Integrated Health Insights.Integrate metagenomics, metabolomics, and transcriptomics data alongside VOC analysis to provide a multidimensional understanding of the gut microbiomes interaction with dietary fiber. This approach would facilitate deeper mechanistic insights and improve the predictive power of microbiota-health associations.

3.     Implement Controlled Environment Validation.Conduct validation studies in standardized, controlled environments to isolate the influence of external VOC sources and environmental factors.

4.     Expand Applications to Precision Medicine.Position the E-Nose as a tool for precision medicine by coupling it with personalized dietary interventions based on individual VOC and microbiome profiles. Additionally, explore its utility in diagnosing and monitoring conditions like inflammatory bowel disease (IBD), colorectal cancer, and metabolic syndromes through specific VOC biomarkers.

5.     Develop Dynamic Longitudinal Models.Use advanced time-series analysis and dynamic modeling to track individual microbiome shifts over extended periods. This would enable the identification of microbiome resilience patterns and long-term impacts of dietary interventions, making the E-Nose a valuable tool for preventive and therapeutic monitoring.

6.     The authors are suggested to compare and discussion some recent works in the ultrafast photonics fields, (i.e. Ultrafast Science, 3, 0006, 2023)

Author Response

We are incredibly grateful to the reviewers and the editor for providing us with constructive and consistent feedback and for giving us the opportunity to revise and improve our manuscript entitled “Use of a Novel Passive E-Nose to Monitor Fermentable Prebiotic Fiber Consumption.

We have made modifications throughout the manuscript, which we believe addresses all the reviewers’ concerns. We have included detailed point-by-point responses to each of the reviewers' and editor’s concerns below. Our responses have been highlighted in blue to facilitate the review process. We have included a revised version of the manuscript with tracked changes (to ease the review process by the reviewers) and one without tracked changes. We believe that our manuscript has been significantly improved by the reviewers’ constructive comments.

Reviewer #3

Reviewer General Comment: “This paper investigates a home-based Electronic-Nose (E-Nose) to passively monitor volatile organic compounds (VOCs) emitted with a bowel movement and assessed its validity by correlating the output with prebiotic fiber intake. Here are some suggestions.”

R3.1, 2, 3, 4, 5:

  1. Enhance Sensor Precision and Multi-VOC Profiling. Upgrade the E-Nose with multi-channel sensing technology to distinguish between specific volatile organic compounds (VOCs) associated with diverse microbial metabolic pathways. This would allow precise correlation of VOC profiles with microbial taxa and functional pathways, thus increasing the resolution of microbiota monitoring.
  2. Incorporate Omics Data for Integrated Health Insights. Integrate metagenomics, metabolomics, and transcriptomics data alongside VOC analysis to provide a multidimensional understanding of the gut microbiome’s interaction with dietary fiber. This approach would facilitate deeper mechanistic insights and improve the predictive power of microbiota-health associations.
  3. Implement Controlled Environment Validation. Conduct validation studies in standardized, controlled environments to isolate the influence of external VOC sources and environmental factors.
  4. Expand Applications to Precision Medicine. Position the E-Nose as a tool for precision medicine by coupling it with personalized dietary interventions based on individual VOC and microbiome profiles. Additionally, explore its utility in diagnosing and monitoring conditions like inflammatory bowel disease (IBD), colorectal cancer, and metabolic syndromes through specific VOC biomarkers.
  5. Develop Dynamic Longitudinal Models. Use advanced time-series analysis and dynamic modeling to track individual microbiome shifts over extended periods. This would enable the identification of microbiome resilience patterns and long-term impacts of dietary interventions, making the E-Nose a valuable tool for preventive and therapeutic monitoring.

Response: Thank you for bringing attention to these five-study design and data assessment tools. We believe these tools should be applied in future experimental research. In the Discussion, we reiterate the significance of the results from this pilot proof-of-concept study and how this study’s framework could help guide the design of future studies (lines 307-313). Future study designs should aim to address the limitations of our pilot study (lines 351-361), prioritize the use of second-generation devices to improve sensor accuracy and multi-VOC profiling, and integrate Omics data to enhance our understanding of the interactions between the gut microbiome and dietary fiber intake, ultimately facilitating a deeper mechanistic insight (lines 362-369).

R3.6. “The authors are suggested to compare and discussion some recent works in the ultrafast photonics fields, (i.e. Ultrafast Science, 3, 0006, 2023)”

Response: Thank you for your suggestion. We have reviewed recent studies related to this field and added a new paragraph to the Discussion to emphasize their significance and highlight these study approaches (lines 371-382).

Reviewer 4 Report

Comments and Suggestions for Authors

I cannot give further deep comments for this submission. In my opinion, it is has no relationship with the design methods for novel sensors. It only used a device to measure the VOCs values and gives a value method for discussing the relationship between the VOCs output with the fiber intake.

Author Response

Please see the attachment. Thank you, Reviewer 4.

Round 2

Reviewer 2 Report

Comments and Suggestions for Authors

The authors have revised the manuscript and appropriately addressed the questions in the comments. I recommend accepting the current version of the manuscript.

Reviewer 4 Report

Comments and Suggestions for Authors

The authors have improved the manuscript and replied to the comments.